# Improving 3D-aware Image Synthesis with A Geometry-aware Discriminator

**Zifan Shi**[1*] **Yinghao Xu**[2*] **Yujun Shen**[3] **Deli Zhao**[3] **Qifeng Chen**[1] **Dit-Yan Yeung**[1]

[1]HKUST [2]CUHK [3]Ant Group

## Abstract

3D-aware image synthesis aims at learning a generative model that can render photo-realistic 2D images while capturing decent underlying 3D shapes. A popular solution is to adopt the generative adversarial network (GAN) and replace the generator with a 3D renderer, where volume rendering with neural radiance field (NeRF) is commonly used. Despite the advancement of synthesis quality, existing methods fail to obtain moderate 3D shapes. We argue that, considering the two-player game in the formulation of GANs, *only making the generator 3D-aware is not enough*. In other words, displacing the generative mechanism only offers the capability, but not the guarantee, of producing 3D-aware images, because the supervision of the generator primarily comes from the discriminator. To address this issue, we propose *GeoD* through learning a geometry-aware discriminator to improve 3D-aware GANs. Concretely, besides differentiating real and fake samples from the 2D image space, the discriminator is additionally asked to derive the geometry information from the inputs, which is then applied as the guidance of the generator. Such a simple yet effective design facilitates learning substantially more accurate 3D shapes. Extensive experiments on various generator architectures and training datasets verify the superiority of GeoD over state-of-the-art alternatives. Moreover, our approach is registered as a general framework such that a more capable discriminator (*i.e.*, with a third task of novel view synthesis beyond domain classification and geometry extraction) can further assist the generator with a better multi-view consistency. Project page can be found here.

## 1 Introduction

3D-aware image synthesis has received growing attention due to its potential in modeling the 3D visual world. Compared to 2D generation [1, 13–16, 28], which primarily focuses on the image quality and diversity, 3D-aware generation is expected to also learn an accurate 3D shape underlying the synthesized image. However, existing generative models, such as the popular generative adversarial network (GAN) [8], fail to capture the 3D information from 2D images, as they produce an image based on 2D representations only. To bridge this gap, recent studies [2, 3, 5, 9, 23–25, 29, 30, 34, 36] seek help from 3D rendering techniques, like neural radiance field (NeRF) [19], and propose to replace the generator in a GAN with a 3D renderer. Such a replacement enables explicit control of viewpoints when generating an image, benefiting from the 3D awareness of the generator.

Many attempts have been made to improve 3D-aware GANs from the generator side, with either better representations [3, 5, 24, 25, 30, 34] or novel architectures [2, 9, 23, 29, 36]. Despite the advancement of 2D synthesis quality, they still suffer from unsatisfactory 3D shapes, which appear as flatten, noisy, or uneven surfaces as shown in Fig. 1a. We argue that, given the generator-discriminator competition in training GANs, *only making the generator 3D-aware is not enough*. Specifically, the generator

---

* Work was done during an internship under Ant Group.

36th Conference on Neural Information Processing Systems (NeurIPS 2022).

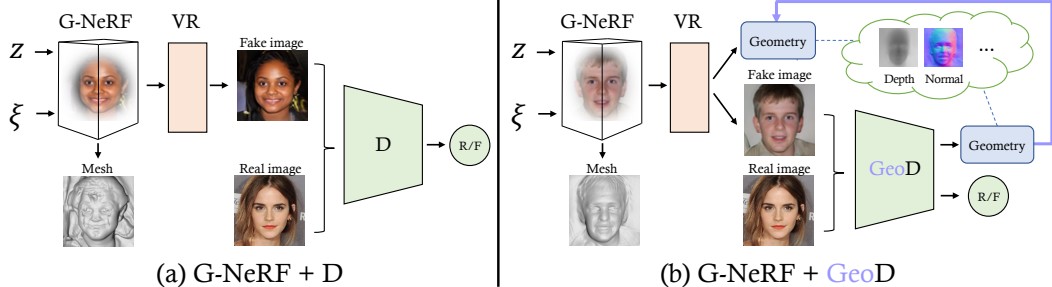

(a) G-NeRF + D            (b) G-NeRF + GeoD

Figure 1: **Concept comparison** between (a) existing 3D-aware GANs where *only* the generator is made 3D-aware with the help of NeRF [19], and (b) our *GeoD* where the discriminator supervises the generator with the extracted geometry. Such an additional task (*i.e.*, geometry extraction) substantially improved 3D-aware GANs with more accurate 3D shapes (see meshes). Here, **z** and $\xi$ denote a sampled latent code and camera pose, respectively, while "VR" stands for volume rendering [19].

obtains supervision (*i.e.*, gradients) mostly from the discriminator, however, the discriminator remains differentiating real and fake images from the 2D image space, as shown in Fig. 1a. Under such a case, even though the generator is potentially capable of learning 3D shapes, it is still exposed to the risk of shape-appearance ambiguity (*i.e.*, well rendered 2D appearance does *not necessarily* imply a decent 3D shape) because of the lack of adequate 3D guidance.

To tackle this problem, we propose a geometry-aware discriminator, termed as *GeoD*, which advances 3D-aware GANs from the discriminator side. As shown in Fig. 1b, besides the basic task of real/fake domain classification, we assign the discriminator an additional task of geometry extraction. For one thing, the classification branch remains the same as that in conventional GANs, and helps the generator improve the realness of the syntheses from the 2D space. For another, the geometry branch targets deriving the shape-related information, such as depth and normal, from a given image. Such information is employed as an extra signal to supervise the generator for a fair 3D shape. Here, the geometry branch is unsupervisedly learned on real data only and jointly updated with the generator, forming a co-evolution manner.

We evaluate our approach on various 3D-aware GAN variants, including $\pi$-GAN [2], StyleNeRF [9], and VolumeGAN [34], to verify its generalization ability. Both qualitative and quantitative results on a range of datasets, including FFHQ (human faces) [14], AFHQ cat (animal faces) [4], and LSUN bedroom (indoor scenes) [35], validate that GeoD outperforms the baselines with a far more accurate 3D shape *without* sacrificing the image quality. Furthermore, we show that our framework is general such that it can be applied for other 3D-related improvements. In particular, when assigned with a third task of novel view synthesis, the discriminator can urge the generator to learn better multi-view consistency, demonstrating our motivation of *making the discriminator 3D-aware*.

## 2 Related work

**3D-aware image synthesis.** 3D-aware image synthesis has received lots of attention recently. The key difference from 2D GANs is the use of 3D representation in the generator. VON [37] and HoloGAN [22] use the voxelized 3D representations to perform 3D-aware image synthesis, but these methods suffer from the 3D inconsistency due to the lack of underlying geometry. [2, 5, 24, 29, 30] introduce NeRF [19] to synthesize 3D-aware images, and [3, 9, 23, 34, 36] propose to render low-resolution feature maps with NeRF and then leverage powerful 2D CNNs to generate high-fidelity images. The underlying shapes of these approaches, however, are noisy and imprecise. To tackle the inaccurate shape, ShadeGAN [25] proposes to introduce the albedo field instead of the radiance field, and adds a lighting constraint by explicitly modeling the illumination process. However, the discriminators of these approaches still distinguish images from 2D space, which is risky to bring the shape-appearance ambiguity due to the lack of 3D supervision. Our main goal is to improve 3D geometry from the discriminator's perspective by introducing geometry-aware discrimination to supervise the 3D-aware generator. It is worth noting that our geometry-aware discriminator can be easily applied to the approaches built upon the neural radiance field.

**Geometry extraction.** Extracting geometries from images has been a long-standing problem. As in most cases only monocular data is available to train a generative model, we only discuss geometry extraction from monocular images here. A traditional way is to train a network with multi-view data or labeled geometries in a supervised manner, such as depth estimation [7, 26, 27] and inverse rendering [11, 17, 18, 32]. By introducing differentiable renderers, recent work [33] starts extracting geometries from monocular images unsupervisedly. Concretely, the network first decomposes an image into several geometry factors (*e.g.*, albedo, depth, and lighting), and a differentiable renderer then reconstructs the image following physical models. However, such methods require assumptions, such as symmetry and shading, and thus they are only applicable on object images. All these tasks and models can be integrated into the discriminator for geometry extraction.

## 3 Method

### 3.1 Preliminaries

**Generative neural radiance field.** The neural radiance field (NeRF) [19] adopts a continuous function, $F(\mathbf{x}, \mathbf{d}) = (\mathbf{c}, \sigma)$, to learn the RGB color $\mathbf{c} \in \mathbb{R}$ and volume density $\sigma \in \mathbb{R}$ given the 3D coordinate $\mathbf{x} \in \mathbb{R}^3$ and the viewing direction $\mathbf{d} \in \mathbb{S}^2$. $F(\cdot, \cdot)$ is typically parameterized with a multi-layer perceptron (MLP). To render an image under the given camera pose, the pixel color is obtained via volume rendering along the corresponding ray with near and far bounds. Owing to NeRF's strong performance in 3D reconstruction and novel view synthesis, recent attempts on 3D-aware image synthesis propose to introduce the generative neural radiance field (G-NeRF) [29], $G(\mathbf{x}, \mathbf{d}, \mathbf{z}) = (\mathbf{c}, \sigma)$, to the generator. It improves the 3D consistency of synthesized images across different camera views, due to the underlying geometry encoded in G-NeRF.

**Geometry extraction from monocular images.** Geometry extraction from single view attempts to predict 3D information, such as depth, normal, and albedo, from monocular photographs. Unsupervised extraction is made possible by differentiable rendering, which enables the gradient calculation of 3D objects through images during the rendering process. It can be formulated into an autoencoding process based on differentiable rendering to learn the geometry factors in an unsupervised manner. The geometry encoder $\Phi(\cdot)$ maps the image into geometry factors, *e.g.,* depth, normal, albedo, and lighting. The decomposed factors are then used to reconstruct the input image through a differentiable renderer.

### 3.2 Geometry-aware discrimination

Recall that we introduce a geometry branch to the discriminator to provide 3D supervision signals for the generator. In this part, we present how the geometry branch is incorporated to build a geometry-aware discriminator for improving 3D-aware image synthesis.

**Learning a geometry-aware discriminator with real images.** Recent attempts on 3D-aware GANs inherit a 2D discriminator which only distinguishes images in 2D space, suffering from the limited 3D supervision for the generator. To make the maximum use of the 3D information encoded in the 2D images and prevent the discriminator from distinguishing images without awareness of geometry, we assign an geometry extraction task beyond bi-class domain classification, which is to derive the geometry information from the given real image $\mathbf{I}_r$. To achieve this, we introduce a geometry branch $\Phi(\cdot)$ and supervise the discriminator with an extra autoencoding training objective:

$$\hat{\mathbf{I}}_r = \Psi(\Phi(\mathbf{I}_r)), \tag{1}$$

$$\mathcal{L}_{geo}^r = l_r(\mathbf{I}_r, \hat{\mathbf{I}}_r), \tag{2}$$

where $\Psi(\cdot)$ represents a differentiable renderer, and $l_r$ is a function that measures image distance.

**Performing geometry-aware discrimination on fake images.** As discussed above, the 3D-aware generator built upon G-NeRF [2, 3, 5, 9, 23–25, 29, 30, 34, 36] can synthesize consistent 3D-aware images owing to the underlying geometry encoded in G-NeRF rather than the supervision signal from the discriminator. In such a case, the underlying geometry cannot be accurate despite the good quality of the synthesized 2D images. To alleviate the issue of the imperfect shape encoded in G-NeRF, we leverage the intrinsic geometry characteristics of a single image and give geometry-aware supervision to the generator. Concretely, we first ask the discriminator to factorize the geometry information

of synthesized images. And then the geometry $\mathbf{k}$ extracted by $\Phi(\cdot)$ is regarded as a pseudo label to supervise the corresponding geometry $\hat{\mathbf{k}}$ encoded in G-NeRF in a loop-back manner. The image generation process is formulated as $\mathbf{I}_f = G(\mathbf{z}, \xi)$, where $\mathbf{z}$ and $\xi$ are the sampled latent code and camera pose. An extra training objective is included to perform geometry-aware discrimination:

$$\mathbf{k} = \Phi(\mathbf{I}_f), \tag{3}$$

$$\hat{\mathbf{k}} = G_k(\mathbf{z}, \xi), \tag{4}$$

$$\mathcal{L}_{geo}^f = l_f(\mathbf{k}, \hat{\mathbf{k}}), \tag{5}$$

where $\mathbf{I}_f$ is the fake image synthesized by the generator, and $l_f$ denotes a loss function that measures the difference between the geometries. $G_k$ represents geometry extraction from G-NeRF.

**Loss functions.** To summarize, with the purpose of making the discriminator geometry-aware, the generator and the discriminator are jointly trained with:

$$\min_{\boldsymbol{\theta}_G} \mathcal{L}_G = -\mathbb{E}_{\mathbf{z} \sim p_z, \xi \sim p_\xi}[f(D(G(\mathbf{z}, \xi)))] + \lambda_{geo}^f \mathcal{L}_{geo}^f, \tag{6}$$

$$\min_{\boldsymbol{\theta}_D} \mathcal{L}_D = -\mathbb{E}_{\mathbf{I}_r \sim p_r}[f(D(\mathbf{I}_r))] + \mathbb{E}_{\mathbf{z} \sim p_z, \xi \sim p_\xi}[f(D(G(\mathbf{z}, \xi)))]$$

$$+ \lambda \mathbb{E}_{\mathbf{I}_r \sim p_r}\left[\|\nabla_{\mathbf{I}_r} D(\mathbf{I}_r)\|_2^2\right] + \lambda_{geo}^r \mathcal{L}_{geo}^r, \tag{7}$$

where $f(t) = -\log(1 + \exp(-t))$ [20] is the negated logistic loss, and the third term in Eq. (7) is the gradient penalty. $\lambda$, $\lambda_{geo}^f$ and $\lambda_{geo}^r$ represent weights for different loss terms. $\boldsymbol{\theta}_G$ and $\boldsymbol{\theta}_D$ denote the parameters of the generator and GeoD, respectively.

### 3.3 Implementation details

**Geometry branch in GeoD.** Considering the complexity of contents in the image, we employ different architectures of geometry branch for objects and scenes. The geometry branch for objects is built upon Unsup3d [33] and trained unsupervisedly, with the assumption that objects are symmetric. In this branch, the encoder decomposes the input image $\mathbf{I}$ into six factors $\Phi(\mathbf{I}) = (\mathbf{d}, \mathbf{n}, \mathbf{a}, \mathbf{l}, \mathbf{v}, \mathbf{c})$, including a depth map $\mathbf{d}$, a normal map $\mathbf{n}$, an albedo map $\mathbf{a}$, a global lighting direction $\mathbf{l}$, a viewpoint $\mathbf{v}$, and confidence maps $\mathbf{c}$. All factors except the viewpoint are estimated under the canonical view. During the reconstruction process, an image $\mathbf{I}_c$ under the canonical view will first be reconstructed from $\mathbf{d}, \mathbf{n}, \mathbf{a}$, and $\mathbf{l}$ using Lambertian shading model. Then, a reprojection operator warps $\mathbf{I}_c$ from the canonical view to viewpoint $\mathbf{v}$ given the depth map $\mathbf{d}$, which gives us a reconstructed image $\hat{\mathbf{I}}$. Considering the symmetry assumption, a second reconstructed image $\hat{\mathbf{I}}'$ is obtained from flipped depth and albedo. To weaken the influence of asymmetric parts, confidence maps, which indicate the pixel-wise symmetry score, are used as the weighting factors of pixel-wise reconstruction loss. The reconstruction loss for geometry branch of objects on real data is thus given by:

$$\mathcal{L}_{geo}^{r,obj} = l_1(\mathbf{I}, \hat{\mathbf{I}}; \mathbf{c}) + \lambda_{flip} l_1(\mathbf{I}, \hat{\mathbf{I}}'; \mathbf{c}) + l_p(\mathbf{I}, \hat{\mathbf{I}}; \mathbf{c}) + \lambda_{flip} l_p(\mathbf{I}, \hat{\mathbf{I}}'; \mathbf{c}), \tag{8}$$

where $l_1$ and $l_p$ denote the pixel-wise $l_1$ loss and the perceptual loss [12], respectively. $\lambda_{flip}$ represents the weighting factor of loss term. The geometry (*e.g.*, normal) used for generator supervision is obtained by warping the geometry under the canonical view to the one under viewpoint $\mathbf{v}$.

Geometry branch for scenes is built on Li et al.'s [18]. The network is constructed in a cascade manner with two stages. Each stage contains two encoders. One encoder will output albedo $\mathbf{a}$, normal $\mathbf{n}$, roughness $\mathbf{r}$ and depth $\mathbf{d}$, while the other one aims at predicting spatially-varying lighting $\mathbf{l}$. The following differentiable rendering layer then renders the diffuse image $\mathbf{I}_d$ and the specular image $\mathbf{I}_s$ based on these outputs. $\mathbf{a}, \mathbf{n}, \mathbf{r}, \mathbf{d}, \mathbf{I}_d, \mathbf{I}_s$ along with the original image will become the inputs of the encoders in the next stage. Because of the large diversity of geometries, materials, and lighting, ground-truth data for each decomposed component is available for training. Therefore, the training loss for geometry branch of scenes is the sum of $l_2$ loss between each predicted factor $q$ and its ground-truth data $\tilde{q}$:

$$\mathcal{L}_{geo}^{r,sce} = \sum_{q \in \{\mathbf{a}, \mathbf{n}, \mathbf{r}, \mathbf{d}, \mathbf{I}_d, \mathbf{I}_s\}} \lambda_q l_2(q, \tilde{q}), \tag{9}$$

where $l_2$ and $\lambda_q$ denote the pixel-wise $l_2$ loss and the weighting factor for each loss term.

**Geometry extraction from G-NeRF.** As discussed above, the underlying geometry encoding in G-NeRF can be leveraged as an implicit constraint for synthesizing 3D-aware images. Our approach aims at supervising the underlying geometry by GeoD and the normal is chosen as the geometry representation to pass the 3D information during the discrimination process. We first obtain the depth map $\mathbf{d}$ regarding different camera rays by performing volume rendering [19] on the depth-axis:

$$\mathbf{d}(\mathbf{r}) = \sum_{i=1}^{N} T_i(1 - \exp(-\sigma(\mathbf{x}_i)\delta_i))z[\mathbf{x}_i], \tag{10}$$

$$T_i = \exp(-\sum_{j=1}^{i-1} \sigma(\mathbf{x}_j)\delta_j), \tag{11}$$

where $\delta_i = ||\mathbf{x}_{i+1} - \mathbf{x}_i||_2$ is the distance between adjacent sampled points, and $z[\mathbf{x}_i]$ denotes the depth value of each point $\mathbf{x}_i$. Then the tangent map, $\mathbf{t}$, is derived from the depth map along $u, v$ directions [33]. Here, we take the $u$ direction as an example:

$$\mathbf{t}_{u,v}^u = \mathbf{d}_{u+1,v}\boldsymbol{K}^{-1}\boldsymbol{p}_{u+1,v} - \mathbf{d}_{u-1,v}\boldsymbol{K}^{-1}\boldsymbol{p}_{u-1,v}, \tag{12}$$

where $\boldsymbol{p}_{x,y} = (x, y, 1)^T$ is the homogeneous coordinate of image pixel $(x, y)$, and $\boldsymbol{K}$ is the camera intrinsic matrix. And then the normal map $\hat{\mathbf{n}}$ can be obtained by performing the outer product on the tangent map $\hat{\mathbf{n}} = \mathbf{t}_{u,v}^u \otimes \mathbf{t}_{u,v}^v$, followed by the point-wise normalization operation.

## 4 Experiments

### 4.1 Settings

**Datasets.** We evaluate the proposed GeoD on three real-world unstructured datasets, including FFHQ [14], AFHQ cat [4], and LSUN bedroom [35]. FFHQ contains unique 70K high-resolution real images of human faces. All images are aligned and cropped following [14]. AFHQ cat includes around 5K images of various cat faces in different poses. We use the cat face detector to get the landmarks and align images following [14]. Images are then cropped to keep the face in the center. There are about 3M images in the LSUN bedroom. The images are captured in various camera views. We use center-cropping to preprocess the images.

**Baselines.** We compare against three state-of-the-art methods in 3D-aware image synthesis, including $\pi$-GAN [2], StyleNeRF [9] and VolumeGAN [34]. Baselines are either released by the authors or trained with official implementations.

**Training.** We inherit the generators from the baselines and build GeoD on baselines' discriminator. We follow the training protocol of the baselines. For human and cat faces, GeoD is trained from scratch along with the original GAN pipeline. As a result of the complexity of indoor scenes, it is difficult to learn reasonable geometries from monocular images only. Therefore, for GeoD of scenes, we pretrain the geometry branch on synthetic data [18] and NYU dataset [21] following Li et al. [18]. The resolution of the geometry branch in GeoD is $64{\times}64$ for faces and $256{\times}256$ for scenes. Images are resized to satisfy this requirement. The entire training ensures the discriminator to see 25000K real images. Due to the expensive rendering process, $\pi$-GAN is trained with images of resolution $64{\times}64$, and VolumeGAN on LSUN bedroom is trained on $128{\times}128$ images. Other experiments are conducted on the resolution of $256{\times}256$. More details are available in *Supplementary Material*.

**Evaluation metrics.** We use three metrics to evaluate the performance, including Fréchet Inception Distance (FID) [10], Scale-Invariant Depth Error (SIDE) [6], and Reprojection Error (RE) [34]. FID is calculated on 50K real images and 50K fake images rendered with random latent codes and camera views, reflecting the quality and diversity of the generated images. SIDE evaluates the generated geometries' accuracy on 10K images by comparing the ground-truth depth and the depth map rendered from the generator. We adopt the pretrained models, LeMul [11] for faces and DPT [27] for scenes, as the depth estimators to extract the depth map from 2D images as ground truths. Since the resolution of NeRF is different from the image resolution in StyleNeRF and VolumeGAN, we resize the ground-truth depth to NeRF's resolution for comparison. RE evaluates the consistency of synthesized images on 10K pairs. We randomly sample two views, and render the image and its depth map under each view given the same latent code. The depth map will be resized to image size if the resolutions of two do not match. Then, we warp the image from one view to the other view using its depth, and calculate the error between the warped image and the image rendered under that view.

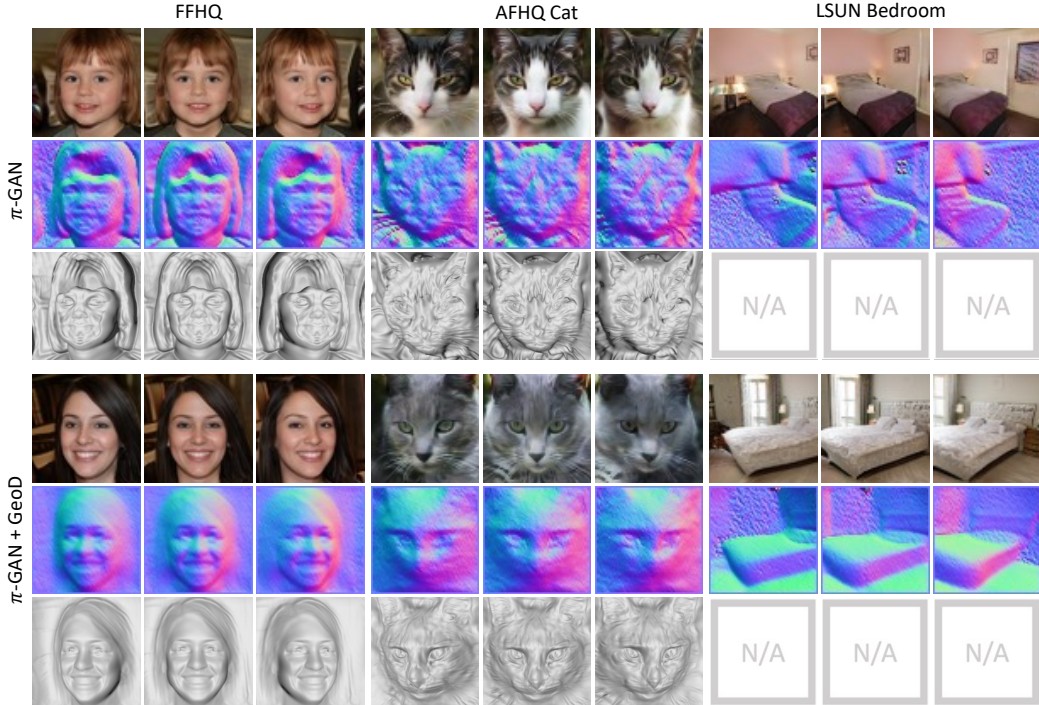

Figure 2: **Qualitative results with $\pi$-GAN [2] as the base model.** From top to bottom: synthesized results, normal maps derived from the generative neural radiance field, and 3D meshes.

Table 1: **Quantitative comparisons** on different GAN architectures and datasets. FID [10] is used to evaluate the 2D synthesis quality, while SIDE [6] and RE [34] helps evaluate the 3D shape. Numbers with * are obtained by pulling the extracted depths to the mean depth plane, because the baselines tend to learn a *flatten* depth either too near or too far, as shown in Figs. 3 and 4.

|  | FFHQ [14] | | | AFHQ Cat [4] | | | LSUN Bedroom [35] | | |
|---|---|---|---|---|---|---|---|---|---|
|  | FID↓ | SIDE↓ | RE↓ | FID↓ | SIDE↓ | RE↓ | FID↓ | SIDE↓ | RE↓ |
| $\pi$-GAN [2] | 8.698 | 0.133 | 0.023 | **9.320** | 0.062 | 0.019 | 15.712 | 0.355 | 0.050 |
| $\pi$-GAN + GeoD | **6.876** | **0.066** | **0.011** | 9.353 | **0.045** | **0.019** | **8.535** | **0.153** | **0.025** |
| StyleNeRF [9] | 8.513 | 0.165 | 0.085 | 5.117 | 0.130 | 0.111 | **8.327** | 0.155 | 0.240* |
| StyleNeRF + GeoD | **7.860** | **0.044** | **0.074** | **4.547** | **0.045** | **0.104** | 9.148 | **0.094** | **0.227** |
| VolumeGAN [34] | 9.598 | 0.084 | 0.168 | **5.136** | 0.053 | 0.128 | **18.107** | 0.451 | 0.098* |
| VolumeGAN + GeoD | **8.391** | **0.039** | **0.061** | 6.475 | **0.038** | **0.043** | 18.796 | **0.113** | **0.050** |

## 4.2 Main results

**2D image quality and 3D shape.** Tab. 1 reports the FID scores evaluated on different models and datasets. With our GeoD, the FID scores are better than or on par with the baselines' in most cases, indicating better image quality and diversity. This is also reflected in the qualitative results in Figs. 2 to 4. For StyleNeRF on bedrooms and VolumeGAN on cats and bedrooms, the FID scores are slightly worse than the baselines'. We conjecture the reason is that these baselines tend to learn a flat shape as shown in Figs. 3 and 4, and thus they reduce the 3D-aware image generation problem to a 2D image generation task, leading to lower FID scores. Our method, however, keeps the image quality comparable with or even better than the baselines while producing correct underlying geometries.

**Analysis on geometries.** We analyze the underlying geometry of the generated images from two perspectives, *i.e.*, the accuracy of the geometry and the consistency across different views, which are reflected by SIDE and RE in Tab. 1. Here, for StyleNeRF [9], SIDE and RE are evaluated only on the foreground objects of FFHQ and AFHQ cat. Lower SIDE and RE values on all datasets and architectures indicate the achievement of better geometries with GeoD. This is also reflected

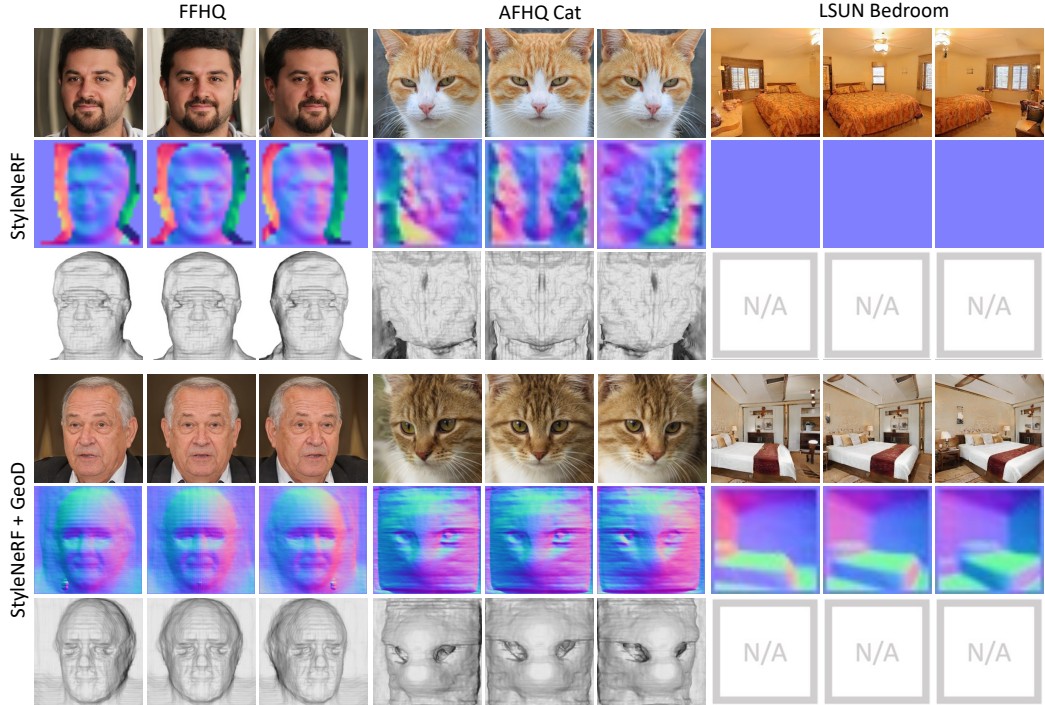

Figure 3: **Qualitative results with StyleNeRF [9] as the base model.** From top to bottom: synthesized results, normal maps derived from the generative neural radiance field, and 3D meshes.

in the qualitative comparisons in Figs. 2 to 4. For each sample, we uniformly sample three views and show the normal map and the mesh at each view. $\pi$-GAN generates noisy shapes on three datasets. With our GeoD, the pits are all gone and the generated shapes are more vivid, resulting in better accuracy and consistency scores. StyleNeRF and VolumeGAN apply NeRF in a lower resolution than the image resolution, and tend to generate flat shapes as shown in Figs. 3 and 4. Especially on AFHQ cat and LSUN bedroom datasets, their generated geometries are nearly or completely flat planes. By using GeoD as the discriminator, the resulting geometry is much more three-dimensional. VolumeGAN combines the volume representation and neural radiance fields for rendering, and therefore the generated shape is finer than that of StyleNeRF. Fig. 5 visualizes the geometries extracted from the 3D-aware generator and GeoD. GeoD successfully extracts vivid geometries and thus is able to guide the generator to produce fine geometries.

### 4.3 Ablation study

We conduct ablation studies to analyze the effectiveness of GeoD under three settings: trained from scratch, pretrained, and fine-tuned. $\pi$-GAN is chosen as the backbone and all experiments are done on FFHQ 64×64. In the first setting, GeoD is trained together with the generator from scratch. In the second setting, we first pretrain the geometry branch of GeoD on FFHQ dataset until it converges. Then, the geometry branch of GeoD will be frozen during the entire training of GAN. In the third setting, the geometry branch of GeoD will first be trained offline on FFHQ dataset for half of the iterations that it converges. After that, the weights will be used to initialize the geometry branch of GeoD in the training of the whole framework. Results are presented in Tab. 2. Pretrained geometry branch in GeoD provides better supervision of geometry at the beginning, hence the accuracy of the geometry is better. However, too strong geometry supervision from the start may dominate the supervision signal and thus leads to a sub-optimal solution for RGB synthesis. Therefore, the setting that GeoD is trained from scratch achieves better image quality and consistency.

Table 2: **Ablation study** conducted on FFHQ [14] under 64×64 resolution using $\pi$-GAN [2].

|  | FID↓ | SIDE↓ | RE↓ |
|---|---|---|---|
| Trained from scratch | **6.876** | 0.066 | **0.011** |
| Pretrained | 7.221 | **0.057** | 0.015 |
| Fine-tuned | 7.136 | 0.058 | 0.014 |

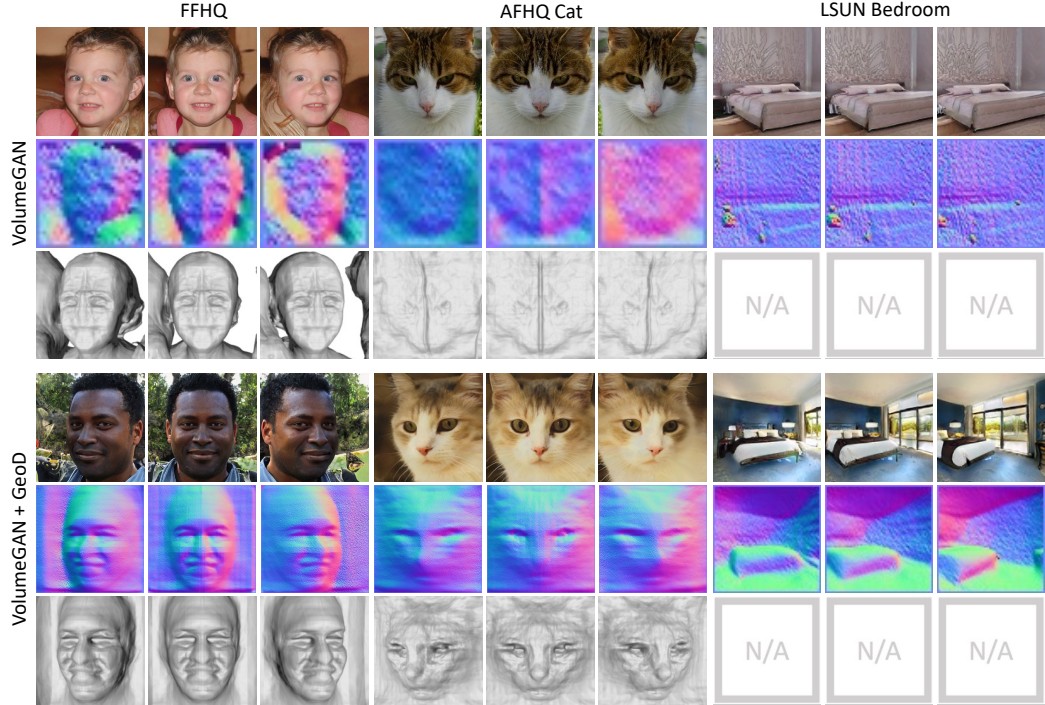

Figure 4: **Qualitative results with VolumeGAN [34] as the base model.** From top to bottom: synthesized results, normal maps derived from the generative neural radiance field, and 3D meshes.

## 4.4 Extension of GeoD on consistency improvement

Besides the underlying geometry, multi-view consistency is also important for 3D-aware image synthesis. In this section, we show that GeoD is a *general* framework such that it is also applicable to improve the multi-view consistency with a third novel view synthesis task. Specially, we ask the generator to synthesize $N$ multi-view images $\{I_i\}_{i=0}^{N-1}$, where $\{I_i\}_{i=0}^{N-2}$ are treated as the source images. These source images are then used to reconstruct an image, $I_{N-1}^{rec}$, which shares the same view as $I_{N-1}$. The difference between $I_{N-1}^{rec}$ and $I_{N-1}$ reflects

Table 3: **Extension of GeoD on improving multi-view consistency.** For this purpose, we equip the discriminator with a third branch, learned with the task of *novel view synthesis*. Such a design improves the multi-view consistency (RE) *without* sacrificing 2D image quality (FID) and 3D shape (SIDE).

|  | FID↓ | SIDE↓ | RE↓ |
|---|---|---|---|
| StyleNeRF | **8.327** | 0.155 | 0.240 |
| StyleNeRF + GeoD $_{con}$ | 8.800 | **0.153** | **0.217** |
| StyleNeRF + GeoD $_{geo}$ | **9.148** | 0.094 | 0.227 |
| StyleNeRF + GeoD $_{geo+con}$ | 9.222 | **0.090** | **0.204** |
| VolumeGAN | **18.107** | 0.451 | 0.098 |
| VolumeGAN + GeoD $_{con}$ | 18.140 | **0.382** | **0.093** |
| VolumeGAN + GeoD $_{geo}$ | **18.796** | 0.113 | 0.050 |
| VolumeGAN + GeoD $_{geo+con}$ | 18.861 | **0.112** | **0.042** |

the degree of consistency and in return guides the generator to be more consistent across different views. We instantiate the consistency branch with IBRNet [31], a 3D reconstruction model pretrained on multiple scene reconstruction datasets. In the experiment, we set $N$ to be 6 and work on LSUN bedroom dataset. We choose StyleNeRF and VolumeGAN as baselines, because they face more severe inconsistency issues as convolutional layers and upsampling operators are introduced to compensate the low-resolution NeRF inside for high-resolution synthesis. Tab. 3 reports the evaluation on models with or without the consistency branch. With consistency branch, FID scores are on par with baselines' or experience a slightly degradation. While the geometries become better given the supervision from the consistency branch, which is indicated by the SIDE metric. The consistency of the generated images is also improved. The results demonstrate the necessity of introducing a consistency branch in the discriminator to enhance the multi-view consistency of the generator.

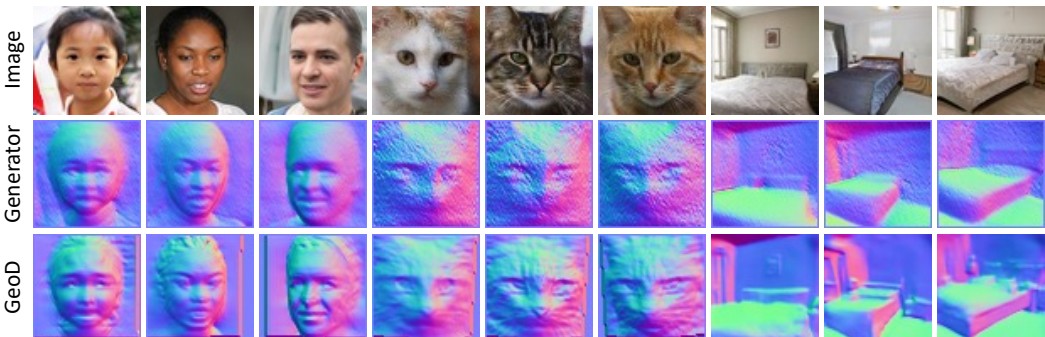

Figure 5: **Qualitative analysis on the geometries** learned by GeoD. From top to bottom: synthesized images, normal maps extracted from the generator, and normal maps predicted by the geometry branch of our GeoD. We can tell that our *geometry-aware* discriminator indeed supervises the generator with adequate 3D knowledge, resulting in more accurate 3D shapes.

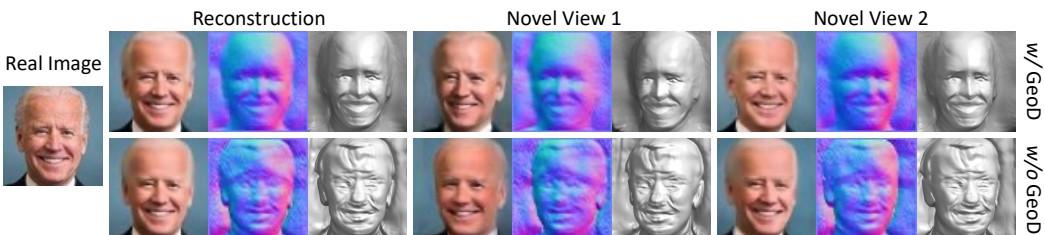

Figure 6: **3D reconstruction and novel view synthesis on real data** with GAN inversion. Thanks to our *geometry-aware* discriminator, the generator in GeoD is capable of learning a more accurate 3D shape and hence reversing it achieves far better shape reconstruction. Meanwhile, on the task of novel view synthesis, which largely relies on the underlying shape, our GeoD presents results with higher fidelity as well as higher similarity to the source image (*e.g.*, novel view 1).

## 4.5 Application of GeoD in GAN inversion

One of the potential applications of GeoD is to help better geometry reconstruction from a real image. To extract the underlying shape from a real image, GAN inversion is performed on the 3D-aware generator following [2]. As shown in Fig. 6, the generator of $\pi$-GAN trained with a 2D discriminator reconstructs geometries with bumpy surfaces, resulting in inconsistent novel view synthesis. With our geometry-aware discriminator, the reconstructed shape becomes smooth and realistic, and the images synthesized under different viewpoints are more consistent with each other.

## 5 Conclusion

In this paper, we propose a new geometry-aware discriminator, *GeoD*, to improve 3D-aware image synthesis. GeoD not only provides supervision on 2D images but also guides the learning of the underlying geometry. With such a discriminator, 3D-aware generators produce high-fidelity images with better underlying geometry. Besides, we further extend our GeoD to improve the mult-view consistency of the generator. Experimental results demonstrate the effectiveness of our proposed discriminator from both the image quality and the geometry perspectives. We hope our work can bring more attention to the research on effective discriminators.

**Discussion.** Despite the high quality achieved in both the 2D image and the geometry, the quality of the generated geometries is influenced by the performance of the geometry branch in GeoD. For example, geometry branch for scenes requires labeled data for training. However, these labels are not available in the current dataset for synthesis. Consequently, training the geometry branch on the same dataset as the generator is impossible, and the one pretrained on other datasets is adopted. The domain gap between the two datasets degrades the quality of supervision signals on geometries, and thus leads to a sub-optimal solution for the generator. A more efficient geometry branch or a novel fine-tuning method can be proposed to alleviate the problem.

## Acknowledgments and Disclosure of Funding

This work has been made possible by a Research Impact Fund project (R6003-21) and an Innovation and Technology Fund project (ITS/004/21FP) funded by the Hong Kong Government.

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
