# Improving 3D-aware Image Synthesis with A Geometry-aware Discriminator – *Supplementary Material* –

**Zifan Shi**[1]    **Yinghao Xu**[2]    **Yujun Shen**[3]    **Deli Zhao**[3]    **Qifeng Chen**[1]    **Dit-Yan Yeung**[1]

[1]HKUST        [2]CUHK        [3]Ant Group

We organize the supplementary material as follows. Sec. A describes the implementation details of baselines. Sec. B introduces the training configurations of our GeoD. Sec. C provides more qualitative comparisons. Sec. D discusses the broader impact of our proposed method. Sec. E analyzes the failure cases of GeoD. Sec. F shows the syntheses under extreme views. Sec. G demonstrates the high-resolution synthesis results.

## A    Baselines

$\pi$**-GAN [1].** We use the official implementation of $\pi$-GAN. Models on all datasets are trained from scratch following the original training pipeline. The training configuration for FFHQ dataset is the same as the one for CelebA provided by the authors except that the number of points sampled along each ray is 12. For AFHQ cat dataset, yaw and pitch angles are uniformly sampled from [-0.3, 0.3] and [-0.155, 0.155], respectively. We sample 12 points along each ray for training. Other settings are the same as those for Cats dataset provided by the authors. For LSUN bedroom dataset, we sample the camera poses from a uniform distribution, with the horizontal range [-$\pi/6$, $\pi/6$]. The pitch angle is fixed to $\pi/2$. The coefficient of the gradient penalty is 10. We set the number of points along each ray to 24. The field of view for FFHQ and AFHQ cat is $12°$, while that for LSUN bedroom is $30°$.

**StyleNeRF [3].** We adopt the official implementation of StyleNeRF. We directly borrow the weights of the model on FFHQ provided by the authors. For the other two datasets, we train the models from scratch. The training configuration for AFHQ cat dataset follows the official one for training on the entire AFHQ dataset. For LSUN bedroom, the background rendering is disabled. We uniformly sample the yaw angle from $[-\pi/6, \pi/6]$ and the pitch angle is set to $\pi/2$. For each ray, the number of points is 24, and the depth range is [0.7, 1.3]. We set the field of view to $30°$.

**VolumeGAN [6].** We use the official implementation provided by the authors. We directly use the pre-trained model on FFHQ for evaluation. We train a model on AFHQ cat dataset from scratch, with the same camera distribution and field of view as those used in StyleNeRF. The number of points per ray is 12. Other hyper-parameters remain the same as the official ones. For LSUN bedroom, we train the model with camera poses sampled from a uniform distribution, ranging from $-\pi/6$ to $\pi/6$. Other settings are kept the same as the official ones.

## B    Training configurations

Considering that the geometry branch of the discriminator may not get well-learned at the start of training, we linearly adjust the loss weight $\lambda_{geo}^{f}$ as

$$\lambda_{geo}^{f} = \lambda_{geo}^{f,start} + (\lambda_{geo}^{f,end} - \lambda_{geo}^{f,start}) * \texttt{clip}(\frac{it - it_{start}}{it_{end} - it_{start}}, 0, 1).$$

We summarize the hyper-parameters of each model in Tab. S1. For geometry branch of objects, we set $\lambda_{flip}$ and $\lambda_p$ to 1 and 0.5 for all models on FFHQ and AFHQ cat. The field of view and depth range

36th Conference on Neural Information Processing Systems (NeurIPS 2022).

Table S1: **Training configurations** under various experimental settings.

| Base model | Dataset | Adjustment | $\lambda_{geo}^{f,start}$ | $it_{start}$ | $it_{end}$ | $\lambda_{geo}^{f,end}$ | $\lambda_{geo}^{r}$ | $\lambda$ |
|---|---|---|---|---|---|---|---|---|
| $\pi$-GAN [1] | FFHQ [4] | ✗ | 1 | - | - | - | 1 | 0.2 |
| | AFHQ cat [2] | ✗ | 0.5 | - | - | - | 1 | 0.2 |
| | LSUN bedroom [7] | ✓ | 0.1 | $20K$ | $60K$ | 1 | 1 | 1 |
| StyleNeRF [3] | FFHQ [4] | ✓ | 0.1 | $200K$ | $1000K$ | 0.5 | 1 | 0.5 |
| | AFHQ cat [2] | ✓ | 0.1 | $200K$ | $1000K$ | 5 | 1 | 0.5 |
| | LSUN bedroom [7] | ✓ | 1 | $200K$ | $2000K$ | 10 | 1 | 0.5 |
| VolumeGAN [6] | FFHQ [4] | ✗ | 1 | - | - | - | 1 | 10 |
| | AFHQ cat [2] | ✓ | 1 | $10K$ | $50K$ | 5 | 1 | 1 |
| | LSUN bedroom [7] | ✓ | 0.1 | $20K$ | $60K$ | 0.5 | 1 | 10 |

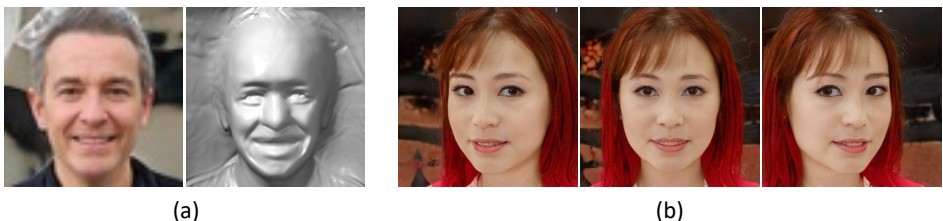

(a)          (b)

Figure S1: **Failure cases.** (a) Coarse shape for fine-grained short hair. (b) Inconsistent eye movement.

of the geometry branch share the same values as those used in the generator. Due to the unstable training, we stop updating the geometry branch of objects on real data after $90K$ iterations for FFHQ dataset, and $50K$ iterations for AFHQ cat dataset. For geometry branch of scenes, the values of $\{\lambda_q\}$ for $q$ in $\{\mathbf{a}, \mathbf{n}, \mathbf{r}, \mathbf{d}, \mathbf{I}_d, \mathbf{I}_s\}$ are $\{1.5, 1.0, 0.5, 0.5, 10, 10\}$. When including the consistency branch (*i.e.*, the third task of novel view synthesis) for training, we adopt the weights trained on $20000K$ images without consistency branch to initialize the network, and then train the model with an additional consistency branch for $5000K$ images. The weighting factor for the consistency loss is 10. All models are trained on 8 NVIDIA V100 GPUs for 2-5 days, depending on the model size.

## C    More results

We provide more qualitative comparisons with $\pi$-GAN [1], StyleNeRF [3], and VolumeGAN [6] in Figs. S4 to S6. Besides the static images under three different views, a demo video is also attached to demonstrate the superiority of our GeoD for continuous 3D control.

## D    Broader impact

Our work can benefit applications in vision and graphics, such as gaming, arts creation, movie production, *etc*. However, as a long-standing problem in the generative fields, generative models can be potentially misused for DeepFake-related applications, *e.g.*, generating fake talking heads in social media. Our work encounters the same problem. Verification cues, such as fingerprints and forensics, are encouraged to add in the generated images or videos to mitigate the issue. Besides, we hope robust DeepFake detection methods can be developed to prevent the misuse of generative models, and our work can also be used to improve DeepFake detection methods.

## E    Failure cases

Our method finds it hard to generate a decent geometry for fine-grained short hair as shown in Fig. S1, which is a result of inadequate backbone [5] used in the geometry branch. We believe with a more carefully designed geometry branch, such a issue can be eliminated. Besides, the inconsistent movement of eye ball remains unsolved.

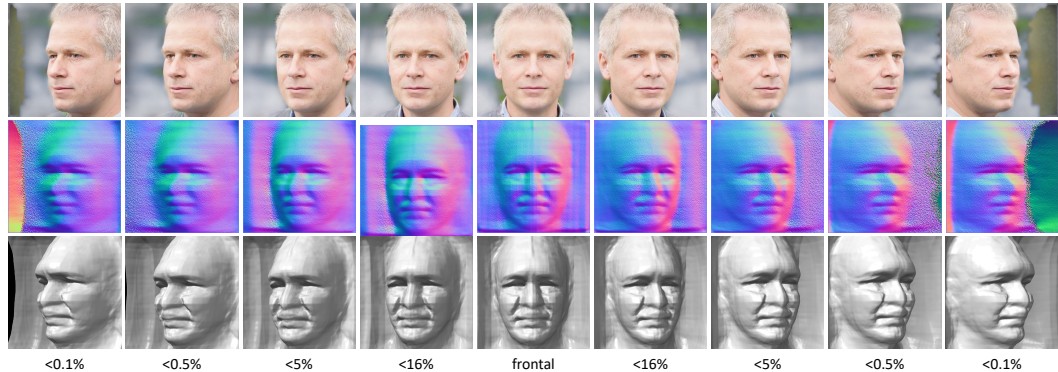



| <0.1% | <0.5% | <5% | <16% | frontal | <16% | <5% | <0.5% | <0.1% |



Figure S2: **Syntheses under extreme views.** $< X\%$ denotes less than $X$ percent of training cases are trained under that pose.

Table S2: Comparison between StyleNeRF and StyleNeRF+GeoD on FFHQ $512^2$ and $1024^2$.

| | FFHQ $512^2$ | | | FFHQ $1024^2$ | | |
|---|---|---|---|---|---|---|
| | FID↓ | SIDE↓ | RE↓ | FID↓ | SIDE↓ | RE↓ |
| StyleNeRF | 7.8 | 0.178 | 0.119 | 8.1 | 0.181 | 0.121 |
| StyleNeRF + GeoD | **7.1** | **0.050** | **0.085** | **8.0** | **0.058** | **0.082** |

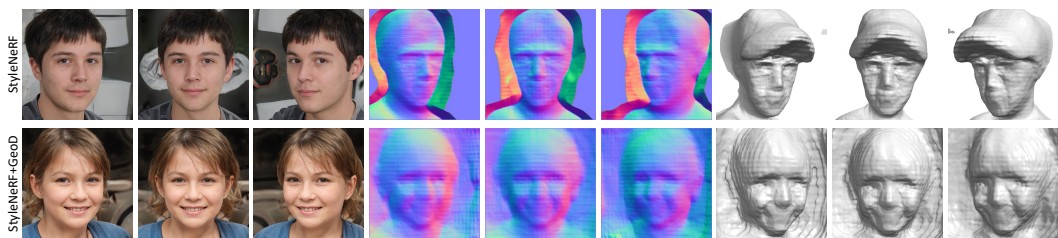

Figure S3: **High-resolution image generation on FFHQ** $1024^2$**.** In each row, we show the image, normal map, and mesh synthesized under three different views. The first row is generated from StyleNeRF [3] while the second row is synthesized from StyleNeRF+GeoD. Though StyleNeRF generates images with high quality, the underlying geometry is flawed (*e.g.*, the forehead hair). With GeoD, the quality of the geometry improves.

## F    Syntheses under extreme views

We synthesize images under extreme views using the generator from VolumeGAN+GeoD in Fig. S2. Even under extreme views, our method can generate consistent images with correct underlying shapes.

## G    High-resolution image generation

We train StyleNeRF [3] with our proposed GeoD on FFHQ $512^2$ and $1024^2$ to show the ability of our method on high-resolution image generation. Qualitative result is available in Fig. S3 and quantitative result is shown in Tab. S2. With GeoD, the generated image along with its underlying shape get better both qualitatively and quantitatively.

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

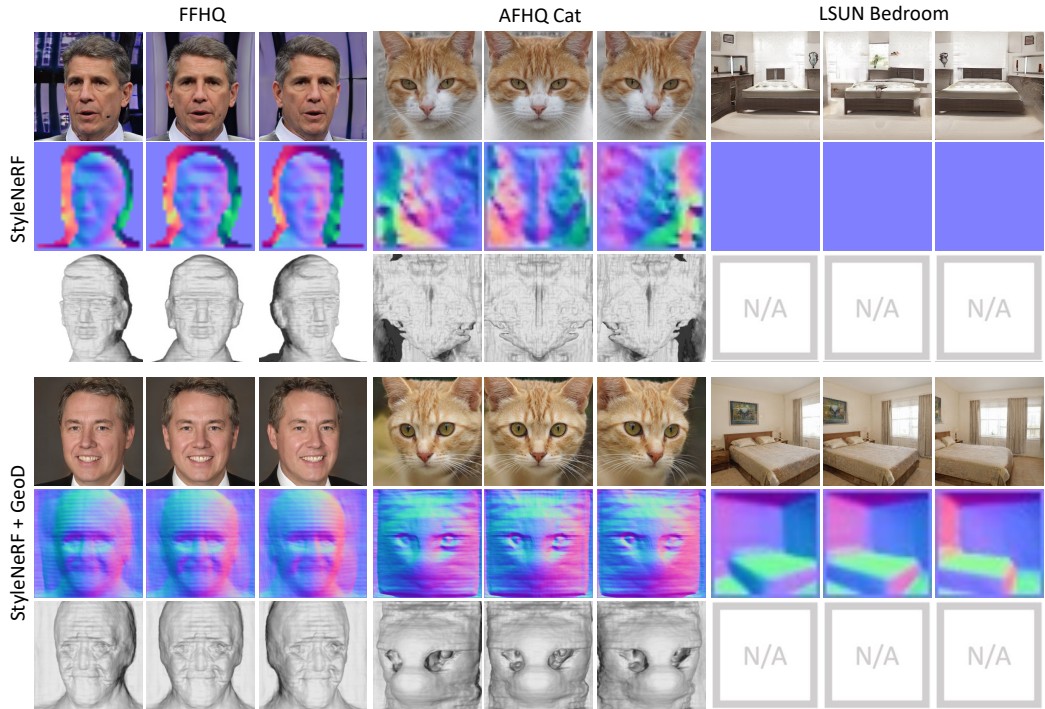

Figure S5: **Qualitative results with StyleNeRF [3] as the base model.** From top to bottom: synthesized results, normal maps derived from the generative neural radiance field, and 3D meshes.

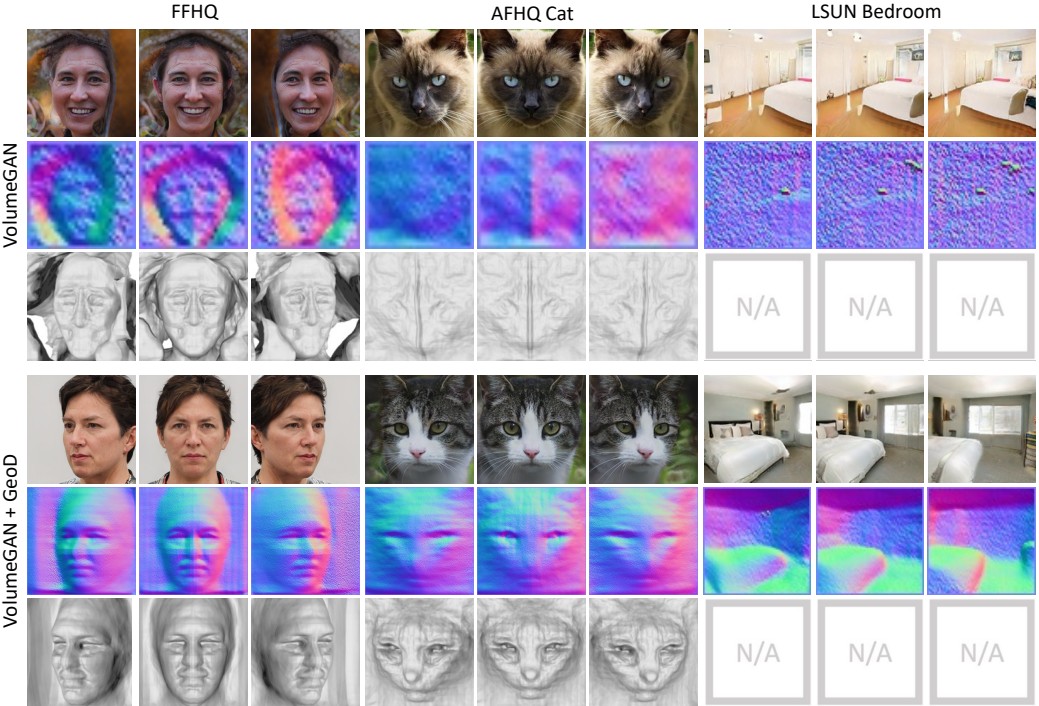

Figure S6: **Qualitative results with VolumeGAN [6] as the base model.** From top to bottom: synthesized results, normal maps derived from the generative neural radiance field, and 3D meshes.