# OpenReview forum: "Improving 3D-aware Image Synthesis with A Geometry-aware Discriminator"
_NeurIPS.cc/2022/Conference — NeurIPS 2022 Accept_

### Official Review · Reviewer_PMt3 · 2022-07-10

**Rating:** 6
**Confidence:** 3
**Soundness:** 3 good
**Presentation:** 2 fair
**Contribution:** 3 good

**Summary:**

The current discriminators of 3D-aware GANs only discriminate between real and fake. This paper proposes learning a geometry-aware discriminator to improve 3D-aware GANs. In particular, the discriminator is additionally asked to derive the normal from the inputs, which is then applied as supervision for the generator. The author also introduces a consistency branch for the discriminator and claims such a design could improve multi-view consistency. Experimental results show the effectiveness of the proposed discriminator in both image and geometry quality.

**Questions:**

Please see the weaknesses.

**Limitations:**

Yes, they did.

**Strengths And Weaknesses:**

Strengths:

The problem studied in the paper is important and the method is a possible direction for improving the view consistency of 3D GANs.

The proposed method shows an improvement in geometry quality.


Weaknesses:

1. The discriminator learns to regress 3D attributes with an autoencoding training objective for real images. The discriminator then estimates the normals of fake images, which are used as pseudo labels to supervise the generator. Would it be better to use a separate network as the 3D attributes regressor?

2. In equation 12, the author should provide enough details to understand the equation. What do all the terms mean? For example, explain the meaning of d_{u+1} and d_{u-1}, and detail the elements of the intrinsic matrix K.

3. The authors did not verify the effectiveness of GeoD on high-resolution experiments (e.g., at 1024x1024 resolution). The qualitative results are not very impressive. As shown in the demo video, the generated images are blurry. The ablation study was limited to 64x64 resolution images, and the results were not very convincing. The same goes for the GAN inversion experiment. I recommend the author demonstrate the effectiveness of GeoD on high-resolution images.

4. The paper structure needs to be adjusted. For example, Section 4.4 proposes a consistency branch, which should be moved to the methods section.

5. In Figures 2 and 3, why is the bedroom mesh not shown?

6. The approach seems to be ad-hoc. For example, the author employs different architectures for the geometry branch for objects and scenes, limiting the scalability of the method.

---

> ### Author Response · Authors · 2022-08-02
> **Response to Reviewer PMt3**
>
> **Q1: Would it be better to use a separate network as the 3D attributes regressor?**
>
> What we want to deliver in this work is that *only making the generator in a GAN 3D aware is not enough for 3D-aware image synthesis*. We empirically verify that, through making the discriminator 3D aware as well, the generator could gain better supervision and hence learn more accurate 3D shape. In our framework, we do not make any assumption on the newly introduced branch. In fact,
>
> - It can share the backbone with the original branch, or isolated as a separate network (the IBRNet in Sec. 4.4 is completely independent of the main branch).
>
> - It can load pre-trained weights, or learned from scratch jointly with the generator (see Sec. 4.3 and Tab. 2 of the submission)
>
> - It can be used to extract geometry information with inverse rendering (Sec. 3), or used to regularize the multi-view consistency with novel view synthesis (Sec. 4.4), *etc.*
>
> From this perspective, our framework is highly flexible. Investigating the network design of the newly introduced branch or whether it should share the backbone with the primary branch is not our main focus.
>
>
> **Q2: About Eq. (12).**
>
> $\mathbf{d}$ is defined in Eq. (10), denoting the depth. The subscripts u and v denote the coordinates in the image plane, and $\mathbf{d}_{u,v}$ represents the depth value at the coordinate (u, v).
> $\mathbf{K}$ is in the form of $\begin{bmatrix}
> f & 0 & c_u\\\\
> 0 & f & c_v\\\\
> 0 & 0 & 1
> \end{bmatrix}$, where $f$ is the focal length that shares the same field of view as the generator.
> ($c_u$, $c_v$) is the center point of the image plane.
>
>
> **Q3: About high-resolution image generation.**
>
>
> Our work targets providing a better 3D supervision to the generator by making the discriminator 3D aware. Such an idea is *orthogonal and synergistic* to existing high-resolution 3D-aware image synthesis approaches. Following the suggestion, we apply our approach to StyleNeRF to generate $512 \times 512$ and $1024 \times 1024$ images. The quantitative comparisons can be found in the table below, while the qualitative results are included in Sec. G of the revised supplementary material.
>
>
> | Resolution  |    |512||   |  1024 |   |
> |---|---|---|---|---|---|---|
> |    Metric       |   FID    |    SIDE    |   RE  |   FID | SIDE  |   RE  |
> | StyleNeRF  |  7.8 | 0.178  | 0.119  |  8.1  | 0.181  |  0.121  |
> | StyleNeRF + GeoD  |  **7.1**  | **0.050**  | **0.085**  | **8.0**  | **0.058**  | **0.082**  |
>
>
> **Q4: Section 4.4 should be moved to the methods section.**
>
> Thanks for the suggestion, but we still believe Sec. 4.4 should be left as where it is. Our contribution lies in the proposal of a new paradigm from 3D-aware image synthesis, where both the generator and *more importantly* the discriminator should be 3D-aware. Sec. 3 takes 3D geometry into account and results in an instantiation. It is well-organized, self-contained, and easier for the readers to follow. Sec. 4.4 introduces another instantiation regarding the 3D consistency, which is primarily used to verify the generalizability of our framework. We treat it as an extension since the geometry quality is a far more challenging and widely-studied problem in the area of 3D-aware image synthesis.
>
>
> **Q5: About the mesh of bedrooms.**
>
> Depth and normal maps are achieved by integrating the per-point information along rays. Different point distribution may lead to the same value for a ray. The ideal case is that all the points on the surface have full density while the other points have empty density. However, given the complexity of bedrooms, we find it hard to learn an adequate distribution per scene. Consequently it is hard to extract a reasonable mesh using marching cube even though the depth and normal maps are correct. This problem is shared by all existing 3D-aware image synthesis approaches. Adding regularizers on the point distribution (*e.g.*, the sparsity regularizer) may be a potential solution, but is beyond the scope of this work.
>
>
> **Q6: About the scalability of GeoD.**
>
> GeoD is flexible since we do not introduce any constraints on the geometry branch. As stated in **Q1**, the geometry branch could share the backbone with the original discriminator, or get optimized independently from the original branch. In this way, all inverse rendering techniques could be incorporated into our framework with no further effort. Beyond the geometry branch, we have demonstrated in Sec. 4.4 that our framework is also compatible with the consistency branch. In other words, our framework is general, and can be improved together with the development of other 3D-related techniques.

---

> > ### Author Response · Authors · 2022-08-06
> > **Thanks for the review.**
> >
> > Dear Reviewer PMt3,
> >
> > Thanks for your efforts in reviewing our paper. We have added the responses to hopefully resolve your concerns.
> >
> > If you have any further concerns or there is anything unclear to you, feel free to let us know and we are happy to discuss more.
> >
> >
> > Sincerely,
> >
> > Authors

---

> > ### Comment · Reviewer_PMt3 · 2022-08-09
> > **Comment.**
> >
> > The author did not address the weaknesses 6. The author employs different architectures for the geometry branch for objects and scenes. If we train 3D GANs on a new type of dataset such as the human body, we need to redesign the architecture of the geometry branch. This is why I say GeoD is not flexible enough.
> >
> > The location of Section 4.4 is very strange, which was also mentioned by Reviewer 7Jaj. Now that two reviewers have noted this, the author should consider whether it is the article's flaw. Also, the principle of Equation 12 requires a more detailed explanation rather than just listing each element of the camera extrinsic matrix K, which is already well-known.
> >
> > As for the high-resolution results (Figure S3 in Supplementary Materials), the human eyes shift with the camera pose. StyleSDF solves this problem through geometric regularization (Adding Eikonal loss to the generator). However, GeoD still suffers from this problem. It seems that the ability of GeoD to regularize the geometry is relatively weak.
> >
> > Considering the weak scalability of GeoD (geometric branches need to be redesigned for different types of datasets), and the current unsatisfactory generating results, I stick to my current rating.

---

> > > ### Author Response · Authors · 2022-08-09
> > > **Disagreements on your new comments.**
> > >
> > > Thanks for your reply. With all due respect, we would like to point out some **misunderstandings and errors** in your additional comments.
> > >
> > > - "If we train 3D GANs on a new type of dataset such as the human body, we need to redesign the architecture of the geometry branch. This is why I say GeoD is not flexible enough."
> > >
> > >   ***Disagree.*** Instead, this is just where GeoD is flexible. The reason is as follows. Given a new dataset, like human bodies, **your mentioned paper, StyleSDF, or other baselines would still *fail* to learn good geometries**. Besides, they **leave no space** for architecture redesigning, leaving it as a **complete failure**. Instead, we **offer the opportunity** for the redesign, making it possible to learn moderate geometry from challenging datasets, like bedrooms.
> > >
> > >   ***If you are familiar with inverse rendering, which we believe is the truth, you will definitely know there are many techniques we can borrow.***
> > >
> > > - "The location of Section 4.4 is very strange, which was also mentioned by Reviewer 7Jaj."
> > >
> > >   With all due respect, you are ***misrepresenting*** the comments from Reviewer 7Jaj. Reviewer 7Jaj comments our presentation as **"The paper is nicely written and the presentation is very clear."** Reviewer 7Jaj regarded Sec. 4.4 as an extension and we explained its necessity to our manuscript in our response (see **Q2 to Reviewer 7Jaj**). Instead, you regarded Sec. 4.4 as a part of our method, **which is not the truth**.
> > >
> > > - "the principle of Equation 12 requires a more detailed explanation rather than just listing each element of the camera extrinsic matrix K, which is already well-known."
> > >
> > >   ***Disagree.*** In your initial review, you said "In equation 12, the author should provide enough details to understand the equation. *What do all the terms mean*? For example, explain the meaning of $d_{u+1}$ and $d_{u-1}$, and *detail the elements of the intrinsic matrix K*." Now, **you are contradicting yourself** by saying K is already well-known, which is **super weird**.
> > >
> > > - "StyleSDF solves this problem through geometric regularization (Adding Eikonal loss to the generator)"
> > >
> > >   ***Disagree.*** StyleSDF does NOT solve this problem. Please refer to the first row of Figure 9 in its [paper](https://arxiv.org/pdf/2112.11427.pdf) as well as the first demo of the teaser video on its [homepage](https://stylesdf.github.io/), where we can observe obvious eye shift phenomenon. Moreover, adding regularizers to the *generator* is orthogonal to our geometry branch introduced in the *discriminator*. Also, as discussed above, StyleSDF fails to generalize to challenging datasets, like bedrooms.
> > >
> > > - "It seems that the ability of GeoD to regularize the geometry is relatively weak."
> > >
> > >   ***Disagree.*** ***If you are familiar with 3D, which we believe is the truth, you will definitely know that eye shift is not a problem of geometry, but texture instead.*** We believe it is **unfair** to blame our geometry branch for texture inconsistency, which may highly be caused by our baseline methods.
> > >
> > > - "the current unsatisfactory generating results"
> > >
> > >     ***Disagree.*** We have already demonstrated the effectiveness and superiority of GeoD on three backbones.
> > >
> > > **We urge a *fair* review to our submission *with no personal bias*, and would like ACs/PCs and other reviewers to carefully go through the comments from Reviewer PMt3 and our response.**

---

> > > > ### Comment · Reviewer_PMt3 · 2022-08-09
> > > > **Comment**
> > > >
> > > > I want to emphasize two facts：
> > > > First of all, Equ 12 is indeed unclear. I think it is necessary to explain its principle in detail.
> > > > Second, StyleSDF has alleviated the eye shift issue by adding geometric constraints. Please refer to Figure 5 (instead of Figure 9 of reprojection images) in the StyleSDF paper which shows randomly generated samples.
> > > >
> > > > According to my experimental experience, the geometric regularization of StyleSDF is indeed useful for the eye shift problem. However, GeoD does not mitigate this problem at all, as shown by their results (Figure S3 in Supplementary Materials). Therefore, I am a little skeptical about the effect of GeoD.
> > > >
> > > > The review process was double-blind, so I do not have any prejudice against this paper. I spent a lot of time reviewing this paper.  After reading the paper, I just think that the current manuscript needs to be polished. Qualitative results also need to be improved to reach an impressive level (e.g. more high-resolution multi-view images instead of low-resolution blurry images).
> > > >
> > > > The author seems to disagree with all my points. I am starting to doubt myself a bit. Maybe ACs will check their articles carefully.

---

> > > > > ### Author Response · Authors · 2022-08-09
> > > > > **Thanks for your reply.**
> > > > >
> > > > > **Q1. About Eq. (12).**
> > > > >
> > > > > Eq. (12) simply projects a 2D pixel, $\mathbf{p}$, to 3D coordinates, according to the camera intrinsic matrix, $\mathbf{K}$, and the corresponding depth $\mathbf{d}$. This is commonly used in inverse rendering. Please refer to paper [33].
> > > > >
> > > > > [33] Unsupervised Learning of Probably Symmetric Deformable 3D Objects from Images in the Wild. Wu *et al.* CVPR'20.
> > > > >
> > > > > **Q2. About StyleSDF alleviating the eye shift issue.**
> > > > >
> > > > > We have the following comments on this claim.
> > > > >
> > > > > - It is *unfair* to ignore Figure 9 and only focus on Figure 5 to conclude that StyleSDF alleviates the eye shift issue, because qualitative results can be cherry-picked.
> > > > >
> > > > > - The Eikonal loss in StyleSDF is *not* proposed to solve the eye shift issue. Instead, the original StyleSDF paper has already **admitted their limitation** that "the reconstructed geometry for human eyes contain artifacts" (Section H of [paper](https://arxiv.org/pdf/2112.11427.pdf)).
> > > > >
> > > > > - There are **still some more important issues to care about** in 3D-aware image synthesis beyond the eye shift issue, such as the unsatisfying overall geometries learned by most prior approaches.
> > > > >
> > > > > **Q3. "Qualitative results also need to be improved to reach an impressive level (e.g. more high-resolution multi-view images instead of low-resolution blurry images)."**
> > > > >
> > > > > From our side, our results are good enough, **especially from the geometry perspective**. Our GeoD is capable of boosting the performance of $\pi$-GAN, VolumeGAN, and StyleNeRF, *both qualitatively and quantitatively*. Also, we have provided the synthesis results at $256\times256$ resolution in the initial submission, and provided additional results at $512\times512$ and $1024\times1024$ resolutions in the revised version following your suggestion. Recall that increasing the synthesis resolution is *not* under the scope of this work. These experiments on high resolutions are just used to verify the generalizability of our approach.
> > > > >
> > > > > If you have any further concerns, we are glad to address them and have a discussion.

---

> > > > > > ### Comment · Reviewer_PMt3 · 2022-08-09
> > > > > > **Comment**
> > > > > >
> > > > > > Q. There are still some more important issues to care about in 3D-aware image synthesis beyond the eye shift issue, such as the unsatisfying overall geometries learned by most prior approaches.
> > > > > >
> > > > > > I agree with this opinion. But GeoD does not seem to improve geometry significantly. For example, the mesh quality of Figure S3 is worse than StyleSDF (equipped with Eikonal loss) and EG3D (equipped with density regularization).
> > > > > >
> > > > > > Q. Increasing the synthesis resolution is not under the scope of this work. These experiments on high resolutions are just used to verify the generalizability of our approach.
> > > > > >
> > > > > > I disagree with this point. For high-resolution images such as 1024x1024 resolution, it is more difficult to estimate normal maps from 2D coordinates by Equation 12. Besides the computational cost of Equation 12 will also increase. Therefore, it is necessary to verify the effect of GeoD on high-resolution image generation rather than limited to low-resolution blurry images.
> > > > > >
> > > > > > There are many checkerboard artifacts in the normal maps of Figure S3. Is it because the normal maps estimated by Equation 12 are not accurate for high-resolution images?

---

> > > > > > > ### Author Response · Authors · 2022-08-09
> > > > > > > **Discussion.**
> > > > > > >
> > > > > > > Thanks for your reply.
> > > > > > >
> > > > > > > First of all, we would like to reaffirm that **our motivation is to make the discriminator in 3D GANs 3D-aware**, which is **orthogonal** to existing studies on improving the generator. We do not claim regularizing the discriminator is better than regularizing the generator, but want to **bring the idea to the community that the discriminator is also important**.
> > > > > > >
> > > > > > > - You keep comparing GeoD with StyleSDF is *unfair*. A more suitable comparison would be equipping GeoD to the generator in StyleSDF. We do *not* choose StyleSDF as a baseline because **it fails to generate indoor scenes, limiting its scalability**, which seems to also be one of your focus. If you are really interested in the performance of combining StyleSDF with GeoD, we can provide a comparison in the next version, but it takes time for training. Note again, GeoD could work together with $\pi$-GAN, VolumeGAN, StyleNeRF, and potentially StyleSDF, but Eikonal loss in StyleSDF is limited to itself.
> > > > > > >
> > > > > > > - Comparing GeoD with EG3D is far more *unfair*, because EG3D relies on the ground-truth pose as the supervision for training.
> > > > > > >
> > > > > > > Second, we do *not* agree that $256\times256$ are low-resolution blurry images. **Most work conducts experiments at such a resolution** to verify the effectives, such as $\pi$-GAN, VolumeGAN, StyleNeRF.
> > > > > > >
> > > > > > > Third, we notice that you keep mentioning qualitative results. We would like to kindly remind you that we also provide **quantitative results**, which are more reliable, because qualitative results can be cherry-picked. For Fig. S3, we followed the setting of StyleNeRF and trained the neural radiance field on a small resolution of $32\times32$. That is why checkerboard effects appear in both the baseline and ours.
> > > > > > >
> > > > > > > We would like to emphasize again that the key contribution of this work is to bring the community's attention on the discriminator, which is important to be 3D-aware *as well*.

---

### Official Review · Reviewer_mehJ · 2022-07-12

**Rating:** 6
**Confidence:** 4
**Soundness:** 3 good
**Presentation:** 3 good
**Contribution:** 3 good

**Summary:**

This work addresses the task of 3D-aware image synthesis. In particular, the authors present a general formulation called GeoD that can be used to improve the performance of various existing 3D-aware GAN variants wrt to the plausibility of the generated images and the reconstruction quality. Specifically, the authors propose to use the output of the discriminator to supervise the extracted geometry of the generator. They evaluate their framework on various GAN variants such as StyleNeRF and VolumeGAN on 3 datasets: FFHQ, AFHQ cat and LSUN bedrooms and show that their method outperforms existing methods wrt to the FID score, the Scale-Invariant Depth Error (SIDE) and the Reprojection Error (RE).


**Questions:**

1. Information regarding the parametrization of the normal map $\mathbf{n}$, the albedo map $\mathbf{a}$ and the global lighting direction $\mathbf{l}$ are not provided  neither in the main paper nor in the supplementary. This needs to be clarified to ensure that reproducibility. Similarly, how is the confidence maps $\bc$ parametrized?

2. In L151-152, the authors state "One encoder will output albedo $\mathbf{a}$, normal $\mathbf{n}$, roughness $\mathbf{r}$ ...". What is roughness and how is it used? This is not explained in the text.

3. I find it a bit weird that for the case of Figure 2 while the normal maps look noisy but have captured the 3D geometry to some extent the authors did not show the 3D meshes. How do the 3D meshes look like?

4. To generate the results of Figure 5, what generator did the authors use?

5. I think that a pictorial representation of the geometry branch in GeoD both for the case of objects and scenes would significantly facilitate understanding the various details presented in section 3.3.

6. Instead of using the proposed geometry parametrization, I think that using an off-the-shelf depth estimator to extract depth maps. My intuition is that this variant would be easier to train and would lead to better results. Have the authors considered this setup?


**Limitations:**

I think that the limitations of this work are adequately discussed in the main paper.


**Strengths And Weaknesses:**

# Strengths:
------------

1. I think that the idea of having a geometry-aware discriminator that is used to supervise the predicted geometry of the generator is very interesting. Moreover, the authors demonstrate that their model can be used in combination with various existing approaches such as StyleNeRF and $pi$-GAN and significantly boost their performance. This clearly shows the contribution of this work.

2. Based on the provided results it seems that the proposed method outperforms methods with non geometry aware discriminators wrt FID, SIDE and RE scores.

# Weaknesses:
-------------

1. I am wondering why didn't the authors consider EG3D (Efficient Geometry-aware 3D Generative Adversarial Networks by Chen at al.) as a baseline? This method has demonstrated superior results in comparison to $\pi$-GAN for example. Hence, I think EG3D would have been a stronger baseline. In particular for the case of EG3D I am wondering whether replacing their StyleGANV2 discriminator with GeoD would still lead to improved results.

2. A more powerful baseline in terms of reconstruction fidelity that the authors could consider in  their evaluation is StyleSDF. I am wondering whether any of the existing 3D-aware GAN variants with the proposed discriminator would still be better than StyleSDF.

3. I think that another important aspect of the model that is not properly discussed is its efficiency. Without a doubt using any of the existing 3D-aware GAN models where only the generator is 3D-aware would probably require less time to train and probably would be able to also generate scenes in various resolutions faster, since they do not have to account for the geometry-aware discriminator. However, I am wondering how much slower is the proposed method in terms of training epochs and how much time is needed for rendering various resolution size images? I believe this analysis is valuable since it will allow us to see what are the limitations of the proposed method.

4. Another experiment that the authors could consider including is showing synthesized images, normal maps and 3D meshes from camera poses that significantly differ from the training poses. It would be interesting to see whether this formulation yields smoother reconstructions even under these more challenging scenarios.

5. Since the output of the discriminator is used as pseudo-labels to supervise the geometry produced by the generator, I am wondering how prone is the proposed model to local minima. For example, during the first stages of training when the pseudo-labels are quite noisy, is it possible for the model to get stuck? From Table 2, it seems that the "Trained from scratch" variant is the performing the best in terms of image plausibility (FID) and reprojection error (RE). However, this result is a bit unexpected for me, since I would have expected the Pretrained variant to be the best performing one. Any idea why this is the case?

---

> ### Author Response · Authors · 2022-08-02
> **Response to Reviewer mehJ (Part 1)**
>
> **Q1: Why not consider EG3D as a baseline?**
>
>
> EG3D requires \textit{ground-truth camera pose} as the condition in the training of discriminator for training, which is not required in our method. From this perspective, it is unfair to compare EG3D with our approach since EG3D is more like a supervised approach. Furthermore, the code and dataset used in EG3D are not released before the submission deadline.
>
>
> **Q2: Comparison with StyleSDF.**
>
> Following the suggestion, we report the quantitative comparison between $\pi$-GAN+GeoD and StyleSDF. Here, we use SIDE and RE as the metrics to evaluate the 3D geometry quality. Depths are rendered in the resolution of 64 and further resized to 256 for both methods. For fair comparison, we resize the RGB images generated by StyleSDF from resolution 1024 to 256, and use the $\pi$-GAN+GeoD model trained on resolution 64 to renders RGB images in the resolution of 256 for inference. As the table shown below, our GeoD is on par with StyleSDF regarding the geometry quality (similar SIDE), but outperforms StyleSDF in terms of the 3D consistency (RE). Besides, *StyleSDF fails to get a reasonable result on LSUN Bedroom*.
>
> |        |  SIDE  |  RE   |
> |--------|--------|-------|
> |StyleSDF| 0.044  | 0.027 |
> |$\pi$-GAN + GeoD| 0.044 | 0.0057|
>
>
>
>
>
>
> **Q3: Efficiency of GeoD.**
>
> First, our 3D-aware discriminator is only used at the training phase and will *not* affect the rendering time of the generator at the inference stage. Besides, the newly introduced geometry branch could be learned super efficiently. As the table shown below, our approach barely increases the training time compared to baselines.
>
> |           |   $\pi$-GAN  |    StyleNeRF   |   VolumeGAN   |
> |-----------|--------------|----------------|---------------|
> |without GeoD|  0.589      |    0.577       |   0.547       |
> | with GeoD |   0.655      |    0.654       |   0.609       |
>
>
> **Q4: Syntheses under extreme views.**
>
> Thanks. We have included some qualitative results under extreme camera poses in Sec. F in the revised supplementary material. We can tell that our approach could produce good visual results with moderate underlying 3D shape under such a challenging case.
>
>
> **Q5: During the first stages of training when the pseudo-labels are quite noisy, is it possible for the model to get stuck?**
>
> In the beginning, both the image from the generator and the geometry extracted by GeoD are noisy, and thus it is hard for both to get stuck in a local minima. A pre-trained geometry branch gives stronger geometry guidance for the generator at the start, at which stage the primary domain classification branch does not follow up yet. This may lead to a better solution for geometry while a sub-optimal solution for RGB synthesis. We guess that is the reason why FID and RE of "Trained from scratch" are better than those of "Pretrained", while "Pretrained' is better in terms of the geometry metric, SIDE.

---

> > ### Author Response · Authors · 2022-08-02
> > **Response to Reviewer mehJ (Part 2)**
> >
> > **Q6: About the parametrization of the geometry branch.**
> >
> > The depth map $\mathbf{d}$, the albedo map $\mathbf{a}$, and the confidence map $\mathbf{c}$ are generated from separate encoder-decoder structures. The encoder consists of 5 convolutional layers and Leaky ReLU activations. Decoders for $\mathbf{a}$ and $\mathbf{d}$ contain 4 deconvolutional layers and an upsampling operator. A convolutional layer and a ReLU activation are used between each two deconvolutional layers to refine the results. The decoder for $\mathbf{c}$ consists of 5 deconvolutional layers and ReLU activations, followed by a convolutional layer to output the final map. The normal map $\mathbf{n}$ is derived from the depth map using Equation 12. The global lighting direction $\mathbf{l}$ is predicted by an encoder which contains 6 convolutional layers, 5 ReLU activations for the first 4 layers and a tanh activation for the last one. *We will release the code for reproduction.*
> >
> >
> > **Q7: About the roughness.**
> >
> > Roughness describes the characteristic of the surface material, which is quantified by the deviations in the direction of the normal vector of a real surface from its ideal form. Roughness, together with albedo, normal and depth, are concatenated and fed into a network to predict the spatially-varying lighting.
> >
> >
> >
> > **Q8: About the 3D meshes of bedrooms.**
> >
> > Depth and normal maps are achieved by integrating the per-point information along rays. Different point distribution may lead to the same value for a ray. The ideal case is that all the points on the surface have full density while the other points have empty density. However, given the complexity of bedrooms, we find it hard to learn an adequate distribution per scene. Consequently it is hard to extract a reasonable mesh using marching cube even though the depth and normal maps are correct. This problem is shared by all existing 3D-aware image synthesis approaches. Adding regularizers on the point distribution (*e.g.*, the sparsity regularizer) may be a potential solution, but is beyond the scope of this work.
> >
> >
> >
> >
> > **Q9: What generator is used in Fig. 5?**
> >
> > We use the model "$\pi$-GAN+GeoD" for Fig. 5.
> >
> >
> > **Q10: Pictorial presentation of the geometry branch.**
> >
> > Thanks. Since our main contribution is the proposal of a general framework to make the discriminator 3D-aware, we only provide a \textit{conceptual diagram} in Fig. 1. We will consider adding a detailed pictorial representation to better illustrate the implementation.
> >
> >
> > **Q11: Use an off-the-shelf depth estimator to extract depth maps.**
> >
> > Using off-the-shelf depth estimator can be one of the instantiations of GeoD, which is similar to loading pre-trained weights for the geometry branch. You are correct that such a setup would ease the training difficulty, however, it may also face the risk that the domain used to pre-train the depth predictor differs from the domain used to train the generator. Such a domain shift may degrade the geometry supervision. In our opinion, a combined solution would be better, which is to load the pre-trained depth estimator but allow its parameterization to be tuned together with the generator training. Such a solution may take both advantages. Thanks for the suggestion.

---

> > > ### Author Response · Authors · 2022-08-06
> > > **Thanks for the review.**
> > >
> > > Dear Reviewer mehJ,
> > >
> > > Thanks for your efforts in reviewing our paper. We have added the responses to hopefully resolve your concerns.
> > >
> > > If you have any further concerns or there is anything unclear to you, feel free to let us know and we are happy to discuss more.
> > >
> > >
> > > Sincerely,
> > >
> > > Authors

---

### Official Review · Reviewer_7Jaj · 2022-07-14

**Rating:** 6
**Confidence:** 5
**Soundness:** 3 good
**Presentation:** 3 good
**Contribution:** 3 good

**Summary:**

This paper presents an improved discriminator architecture for 3D-aware image synthesis. The authors advocate to enforce consistency between the geometric features of images (depth, surface normals etc) rendered from the generative NeRF as well as predicted by off-the-shelf methods (Unsup3D). Such context can make discrimination between real and fake images easier for the network. Experiments show that the proposed geometry-aware discriminator as an additional building block helps improve several baseline methods on human/cat faces and LSUN bedroom datasets.


**Questions:**

It would be great if the authors could address all the concerns listed above.

**Limitations:**

Yes

**Strengths And Weaknesses:**


Strengths:
- I like the motivation of making the discriminator geometry-aware, which is what prevalent 2D image discriminators are largely lacking. The idea of utilizing off-the-shelf methods is also simple, and elegantly bridges with the framework of generative neural radiance fields.
- The results clearly show that add geometry awareness improves the 3D geometry interpretability of image generation. Although not strictly improving on the FID score for every metric on every backbone/dataset, overall the results show significant improvements both qualitatively and quantitatively.
- The paper is nicely written and the presentation is very clear.

Weaknesses:
- It would be good if the authors could provide some failure case analysis to better understand the limitation of the proposed method.
- I think Section 4.4 seem somewhat extraneous to the paper, as it's not central to the paper, and (in addition to [18] and [33]) introduces yet another building block IBRNet [31] to the system. The authors should also elaborate why such novel view synthesis component is necessary to improve the synthesis quality, as the original proposed model is already a 3D-aware image synthesis method based on NeRF.
- Why is GRAF [29] called G-NeRF throughout the entire paper? I believe G-NeRF is more commonly referred to another paper [A]. If the authors meant a GAN + NeRF framework in general, I would recommend not inventing an abbreviation to avoid confusion.
- The loss function notation in Eq 2 and 5 seems to be inconsistent.

[A] Meng et al. "GNeRF: GAN-based Neural Radiance Field without Posed Camera." ICCV 2021.

---

> ### Author Response · Authors · 2022-08-02
> **Response to Reviewer 7Jaj**
>
> **Q1: Failure cases.**
>
> As discussed in the submission (L294), one major limitation of our approach is that the geometry branch is not capable enough to provide accurate geometry supervision. Consequently, the geometry learned by the generator, even better than the baseline, may still be rough, especially at the hair region. Improved inverse rendering techniques may help alleviate such a problem. As suggested, we have added some failure cases in Sec. E and Fig. S1 in the revised supplementary material.
>
>
> **Q2: The purpose of Sec. 4.4, and why the consistency branch is necessary to improve the synthesis quality?**
>
> Our key idea is the proposal of a new paradigm for 3D-aware image synthesis, which *makes the discriminator 3D-aware as well* to compete with the 3D-aware generator. Considering that the 3D evaluation mainly falls into two folds, *i.e.*, the geometry quality and the 3D consistency. Adding a geometry branch is one instantiation of our idea, which can provide explicit supervision on the generator to improve the quality of the underlying geometry. In Sec. 4.4, we would like to show that our framework can also be used to improve the multi-view consistency, by simply incorporating a *consistency branch*. We add this extension to show the generalizability of our framework.
>
> The newly introduced consistency branch, together with the novel view synthesis task, does not necessarily improve the synthesis quality. Instead, it helps improve the property of multi-view consistency. Concretely, although the generative neural radiance field is primarily designed for 3D-aware image synthesis, recent works (like StyleNeRF [9] and VolumeGAN [34]) introduce CNN on top of NeRF to allow high-resolution image synthesis. The CNN is performed in the 2D space, and hence fail to guarantee the 3D property. Under such a case, our consistency branch could help improve the cross-view consistency, as shown in Tab. 3 in the submission.
>
>
>
> **Q3: About the name of G-NeRF.**
>
> Thanks. To help track the revision, we do not change this term at this stage. But we will revise this term in the next version without using abbreviation.
>
>
> **Q4: The loss function notation in Eq. (2) and (5).**
>
> Thanks. We have revised it in the paper.

---

> > ### Author Response · Authors · 2022-08-06
> > **Thanks for the review.**
> >
> > Dear Reviewer 7Jaj,
> >
> > Thanks for your efforts in reviewing our paper. We have added the responses to hopefully resolve your concerns.
> >
> > If you have any further concerns or there is anything unclear to you, feel free to let us know and we are happy to discuss more.
> >
> > Sincerely,
> >
> > Authors

---

### Official Review · Reviewer_QrG6 · 2022-07-16

**Rating:** 3
**Confidence:** 5
**Soundness:** 2 fair
**Presentation:** 1 poor
**Contribution:** 2 fair

**Summary:**

This paper proposes to let the discriminator estimate the geometry of the scene from the 2D rendered version of the generated 3D scenes to improve the geometry of the generated scenes.
- The discriminator estimates six geometry factors from a 2D rendered scene.
- The six geometry factors = a depth map, a normal map, an albedo map, a global lighting direction, a viewpoint, and confidence maps.
- The geometry factors supervise the generator to match them.
- The GT depth map and its normal map for the discriminator are given by the volume rendering of depths.
- The proposed method improves various 3D-aware GANs including pi-GAN, StyleNeRF and VolumeGAN.
  - 3D shapes are more accurate in the qualitative results.
- Experiments are conducted on FFHQ, AFHQ cat, LSUN bedroom.


**Questions:**

Please refer to the above section.


**Limitations:**

Yes. Necessity of pretrained networks for extra supervision and its domain gap.

**Strengths And Weaknesses:**

### Strengths
- Letting the discriminator learn the geometry is novel.
- Some sources of supervision are self-supervision.

### Weaknesses
- Confusions
  - L137-139 The D estimates six factors: depth, normal, albedo, lighting, viewpoint, and confidence.
  - L93 The G produces color and density.
  - L163-170 describes the procedure for producing depth and normal from the density of a generated scene.
  - Eq(6) The G tries to match the geometries estimated by the D.
  - **How do the other four geometries guide the generator?**
  - L161 The normal of the generated scene supervises the D.
  - L121-127 The D tries to match the geometries extracted by the G.
  - **How do the other four geometries from the D are supervised? The problem is under-constrained by the below objectives.**
  - L140 The geometries reconstruct the image.
  - L143 The symmetry assumption

- Writing
  - Redundant descriptions
    - D estimates six factors: L95, L137, L151
  - L150-159 is difficult to understand.
  - The description of the proposed method can be largely improved. Current structure is very confusing because the description for G/D architecture and G/D objectives are entangled.

- Evaluation
  - Reprojection error is not sound because there are corner cases where the error can be small with wrong shapes: planar shape.
  - I am not sure  whether the same ablation study and GAN inversion on other backbones will lead to the same conclusion. Inferiority of pi-GAN compared to other 3D-aware GANs is well-known in the research community.

---

> ### Author Response · Authors · 2022-08-02
> **Response to Reviewer QrG6**
>
> **Q1: Confusions of the method.**
>
> This should be a *misunderstanding*.
>
> - The summary "The GT depth map and its normal map for the discriminator are given by the volume rendering of depths." is *incorrect*.
>
>   The normal map extracted by the discriminator serve as the GT of the volume rendered depth.
>
> - The summary "L161 The normal of the generated scene supervises the D" is *incorrect*.
>
>   The normal of the generated scene is extracted as the object to optimize, while the optimization target is provided by D. Hence, D supervises G instead of G supervises D.
>
> - The summary "L121-127 The D tries to match the geometries extracted by the G" is *incorrect*.
>
>   D does *not* match the geometries from G. Instead, the geometries from D are used to supervise G.
>
> - How do the other four geometries guide the generator?
>
>   These four geometries are jointly learned with depth and normal by the inverse renderer. Only the normal map is used for the supervision of the generator.
>
> - How do the other four geometries from the D are supervised?
>
>   Recovering the geometry information from 2D images is the goal of inverse rendering. The other four geometries can be *unsupervisedly* learned with the *renderer* (*e.g.*, the Lambertian shading model in L138-142) as the 3D prior, and the reconstruction error as the loss function (*i.e.*, Eq. (8)).
>
> - New summary.
>
>   - The discriminator employs a geometry branch, which recovers the geometry information from 2D images with inverse rendering techniques.
>
>   - The geometry information is also extracted from the generative neural radiance field (*i.e.*., the generator).
>
>   - The geometry from the discriminator is used as the pseudo ground-truth to supervise the geometry from the generator, resulting a better 3D guidance.
>
>   - The other training objectives are exactly the same as those in the traditional GAN training.
>
>
> **Q2: Redundant description.**
>
> *Disagree*.
>
> - L95 provides preliminaries of inverse rendering, and the listed factors are used to tell the readers "what are the geometry information".
>
> - L137 describes the geometry branch of our discriminator, as well as how it extracts the geometries from a 2D image.
>
> - L151 describes the geometry branch of our discriminator for scenes, as well as
> the way to extract the geometries from a 2D image.
>
> Both descriptions are essential, because we would like to use the geometries extracted by the discriminator from the synthesized image to supervise the generator.
>
>
> **Q3: The description for G/D architecture and G/D objectives are entangled.**
>
> *Disagree*. Our method is described with the following order:
>
> - Sec. 3.1 introduces the preliminaries on 3D-aware generator and inverse rendering.
>
> - Sec. 3.2 introduces the geometry branch of our discriminator (which is our main contribution), as well as how it is learned on real images and used to supervise fake images.
>
> - Sec. 3.3 introduces the implementation details of how geometry is extracted by the discriminator and the generator.
>
>
> **Q4: About the reprojection error.**
>
> *Disagree.* We follow VolumeGAN [34] (which serves as one of our baselines) to use reprojection error (RE) as a metric for 3D consistency evaluation. To eliminate the concern with planar shapes, we have also reported scale-invariant depth error (SIDE) in Tab. 1 to evaluate the accuracy of the underlying shape. SIDE value will be extremely high if the generated shape is a planar shape. Therefore, combining SIDE and RE is adequate for evaluating the 3D shape and consistency. It is noteworthy that our approach could *boost the performance regarding both metrics*.
>
>
> **Q5: Ablation study and GAN inversion on other backbones.**
>
> Inversion is just a simple application to verify that our approach could learn a moderate 3D underlying shape from 2D images. This is not our major focus.

---

> > ### Author Response · Authors · 2022-08-06
> > **Thanks for the review.**
> >
> > Dear Reviewer QrG6,
> >
> > Thanks for your efforts in reviewing our paper. We have added the responses to hopefully resolve your concerns.
> >
> > If you have any further concerns or there is anything unclear to you, feel free to let us know and we are happy to discuss more.
> >
> >
> > Sincerely,
> >
> > Authors

---

> > ### Comment · Reviewer_QrG6 · 2022-08-08
> > **Thanks.**
> >
> > ### Confusion
> >
> > How do the other four geometries guide the generator?
> >
> > > [A] These four geometries are jointly learned with depth and normal by the inverse renderer.
> >
> > How do the other four geometries from the D are supervised?
> >
> > > [B] The other four geometries can be unsupervisedly learned with the renderer.
> >
> > > New summary: The discriminator employs a geometry branch, which recovers the geometry information from 2D images with inverse rendering techniques.
> >
> > Then, from where does D receive supervisions for its depth and normal?
> > I thought that the unsupervised learning of the renderer and the discriminator works the same for the depth and normal. [A] and the [new summary] agrees my thought written in [C] because the D recovers the geometry from (generated) 2D images.
> > > [C] "The GT depth map and its normal map for the discriminator are given by the volume rendering of depths." is incorrect.
> >
> > Although I see the the Eq.2 also supervised the D but it is not enough because there are too many degenerate solutions that optimizes Eq.2. Success of a sole training of the D with only Eq.2 and without the G will support the claim.
> >
> > BTW, there is no explanation for the inverse renderer in the proposed method. I guess it is meant to be the differentiable rendering.
> >
> > ### Q2
> > Now I see the difference between the architectures for objects and scenes. Describing the factors of scenes like "same for d, n, and a, and r, Id, Is instead of l, v, and c" would be less confusing. At least please list the common factors in the same order.
> >
> > ### Q3
> > That's the reason why I think they are entangled. They inherently share the same ingredients (Image -> factors) but they are described in three different languages in the three subsections. The descriptions will be far more readable after merging the same ingredients and the contribution (letting the D predict the geometry and letting the G to follow it) will stand out better.
> >
> > ### Q4.
> > Understood. Please include this description in the metrics paragraph.
> >
> > ### Q5-1 ablation of training schemes with pi-GAN is not addressed.
> > My concern is about possible different conclusions with other backbones (stylenerf and volumegan) because pi-GAN has limited performance and the FID, SIDE, and RE all depend on the backbone performance.
> >
> > ### Q5-2 Inversion with pi-GAN is not properly addressed.
> > My concern is about possible different conclusions with other backbones because pi-GAN has limited performance and the shape depend on the backbone performance.

---

> > > ### Author Response · Authors · 2022-08-09
> > > **Response to Reviewer QrG6's further comments**
> > >
> > > Thanks for your valuable suggestions on a neater presentation of our approach. Here, we would like to address your concerns on inverse rendering.
> > >
> > > - Inverse rendering aims to **estimate physical attributes (like geometry and lighting) from image(s)**. There is **no need to have ground-truth** and the inverse rendering process can be learned in a **completely unsupervised manner**.
> > >
> > > - We adopt the inverse rendering techniques which rely on single image, which is introduced in Sec. 3.1 (Line 95). Inverse rendering is one of the methods for monocular geometry extraction and is a *well-defined* task. It can learn geometry factors in an unsupervised manner via an autoencoding process based on differentiable rendering.
> > >
> > > - About your further question "from where does D receive supervisions for its depth and normal?". The geometry branch of D learned depth and normal **from real images only** through image reconstruction (Eq. (2)).
> > >
> > >
> > > Regarding Q5, the main contribution of our paper is the proposal of **a new paradigm** for 3D-aware image synthesis, which makes the discriminator 3D-aware *as well* to compete with the 3D-aware generator. The results in Fig. 2-4 and Tab. 1 can *adequately* show the superior of our proposed paradigm, where ours achieves better underlying geometries on all backbones. Downstream tasks such as GAN inversion do not affect the main contribution and conclusion we get.
> > >
> > > By the way, "because pi-GAN has limited performance" seems inappropriate. For example, the geometries extracted by StyleNeRF and VolumeGAN on AFHQ cat (Fig. 2-4) are not as satisfying as the geometry extracted by pi-GAN.

---

### Comment · Area_Chair_FgHa · 2022-08-09
**reminder for discussion**

Dear reviewers,

Thank you all for providing valuable comments. The authors have provided detailed responses to your comments. Has the response addressed your major concerns?

If you haven't, it would be great if you could reply to the authors’ responses soon as the deadline is approaching (Tues, Aug 9).

Best,

ACs

---

### Meta-Review · Area_Chair_FgHa · 2022-08-29

**Recommendation:** Accept
**Confidence:** Certain

**Metareview:**

This paper presents a new 3d-aware image generative model. Compared to previous methods such as pi-GAN, the paper introduces a new geometry-aware discriminator, which helps the model learn better 3D shapes. Many reviewers found the paper well-written, the idea novel/interesting, and the results convincing. They also raised several concerns regarding the evaluation, baselines, and high-res synthesis. The rebuttal has addressed most of the concerns. The AC agreed with most of the reviewers and recommended accepting the paper.

**Award:**

No

---

### Decision · Program_Chairs · 2022-09-14

Accept